# Mini Trampoline, a New and Promising Way of SCUBA Diving Preconditioning to Reduce Vascular Gas Emboli?

**DOI:** 10.3390/ijerph19095410

**Published:** 2022-04-29

**Authors:** Kate Lambrechts, Peter Germonpré, Joaquim Vandenheede, Manon Delorme, Pierre Lafère, Costantino Balestra

**Affiliations:** 1Environmental, Occupational, Aging (Integrative) Physiology Laboratory, Haute Ecole Bruxelles-Brabant (HE2B), 1180 Brussels, Belgium; klambrechts@he2b.be (K.L.); pgermonpre@gmail.com (P.G.); joaquim.vandenheede@ulb.be (J.V.); manondelorm@gmail.com (M.D.); plafere@he2b.be (P.L.); 2DAN Europe Research Division (Roseto-Brussels), 1160 Brussels, Belgium; 3Centre for Hyperbaric Oxygen Therapy, Queen Astrid Military Hospital, 1120 Brussels, Belgium; 4Motor Sciences Department, Physical Activity Teaching Unit, Université Libre de Bruxelles (ULB), 1050 Brussels, Belgium; 5Laboratoire ORPHY, EA4324, Université de Bretagne Occidentale (UBO), 29238 Brest, France

**Keywords:** decompression disease, adverse effects, VGE, vascular dysfunction, venous gas emboli, human

## Abstract

**Background:** Despite evolution in decompression algorithms, decompression illness is still an issue nowadays. Reducing vascular gas emboli (VGE) production or preserving endothelial function by other means such as diving preconditioning is of great interest. Several methods have been tried, either mechanical, cardiovascular, desaturation aimed or biochemical, with encouraging results. In this study, we tested mini trampoline (MT) as a preconditioning strategy. **Methods:** In total, eight (five females, three males; mean age 36 ± 16 years; body mass index 27.5 ± 7.1 kg/m^2^) healthy, non-smoking, divers participated. Each diver performed two standardized air dives 1 week apart with and without preconditioning, which consisted of ±2 min of MT jumping. All dives were carried out in a pool (NEMO 33, Brussels, Belgium) at a depth of 25 m for 25 min. VGE counting 30 and 60 min post-dive was recorded by echocardiography together with an assessment of endothelial function by flow-mediated dilation (FMD). **Results:** VGE were significantly reduced after MT (control: 3.1 ± 4.9 VGE per heartbeat vs. MT: 0.6 ± 1.1 VGE per heartbeat, *p* = 0.031). Post-dive FMD exhibited a significant decrease in the absence of preconditioning (92.9% ± 7.4 of pre-dive values, *p* = 0.03), as already described. MT preconditioning prevented this FMD decrease (103.3% ± 7.1 of pre-dive values, *p* = 0.30). FMD difference is significant (*p* = 0.03). **Conclusions:** In our experience, MT seems to be a very good preconditioning method to reduce VGE and endothelial changes. It may become the easiest, cheapest and more efficient preconditioning for SCUBA diving.

## 1. Introduction

Although SCUBA (self-contained underwater breathing apparatus) diving procedures have become safer over time, decompression sickness (DCS) remains a risk, that can be life threatening. It is commonly known that SCUBA diving can induce vascular gas emboli (VGE) during and following ascent, due to ambient pressure drop [1]. Although these VGE are considered a key element in the development of DCS, the exact pathophysiological mechanisms linking VGE to DCS are still unclear. Therefore, additional pathological mechanisms contributing to the occurrence of DCS might be involved [2,3]. For example, functional changes in vascular wall, repeatedly observed by impaired flow-mediated dilation (FMD), endothelial microparticles or oxidative stress following the dive have been linked to the presence of VGE [4,5,6,7] but also to decompression ‘stress’ without significant VGE presence [8]. Nonetheless, the exact involvement of each of those mechanisms contributing to DCS remains unclear. It is however generally accepted that the absence of detectable VGE post-dive is correlated with a very low probability of DCS [2,9]. Therefore, preconditioning strategies aiming to reduce VGE production and, if possible, to preserve endothelial function have been the focus of recent experimental studies.

These preconditioning strategies [10] encompass methods such as dietary supplementation [11,12], oxygen breathing, physical activities, sauna exposure or vibrations. Whole-body vibration administered 30 min before the dive seems to be particularly effective. It has been shown that it can drastically reduce post-dive VGE [13,14] with a net effect superior to nitrogen washout by pre-dive oxygen breathing [12]. The underlying mechanism is supposedly linked to the mechanical dislodgement of endovascular and tissue micronuclei that are considered as VGE precursors [15,16]. Indeed, in the absence of preconditioning strategy, deep knee flexions performed by the test subject during standard Doppler or echocardiographic VGE detection is associated with higher bubble release provoked by the muscular contractions and suggests that this action still causes adherent bubbles to be dislodged from their sites before reaching their critical volume of detachment [16]. Therefore, by limiting the availability of micronuclei inside the vascular system before diving, no “seeds” are present to produce bubbles post diving, reducing bubble production despite significant inert gas supersaturation [14]. 

Likewise, several studies have demonstrated that physical activity or sauna exposure, some hours before the dive, could have a cardiovascular-mediated preventive effect on VGE formation [17,18,19,20]. An animal model with rats demonstrated that a single bout of exercise 20 h pre-dive reduces VGE post-dive and also DCS occurrence and related mortality [21]. 

In humans, the role of exercise has been debated. Indeed, depending on timing and intensity, exercise may either increase or decrease VGE production [17,22,23]. Physical exercise produces (variable) cardiovascular effects, together with some degree of ‘mechanical stimulation’. It is therefore hypothesized that every exercise-based preconditioning may be related at least in part to effects of vibration. Whole-body vibration sessions (e.g., on a vibration mattress) elicit a mechanical activation of tissues, without cardiovascular effort. Exposure to a high temperature environment, such as during a sauna session, can be considered a cardiovascular activation without mechanical stimulation, Then, specific vibration preconditioning [13,20] and sauna preconditioning protocols [20,24] were developed to specifically dissociate the mechanical effect from the cardiovascular effect.

For the purpose of this study, we hypothesized that the combination of mechanical and cardiovascular activations (even of short duration) can be as effective to reduce gas nuclei elimination before diving, with concomitant reduction in post-dive VGE. Therefore, both stimuli (mechanical and cardiovascular) were coupled by using the mini trampoline device, which is a cheap and accessible method, even in confined environments such as boats. Finally, as already seen with sauna preconditioning, we also hypothesized a positive effect on endothelial function, that should be preserved and not reduced as usually seen after diving and particularly when using a normal diving mask [25] which was the case in our setting.

## 2. Materials and Methods

### 2.1. Population 

After explaining the potential risk and obtention of full written informed consent, 8 (5 females and 3 males; mean age 36 ± 16 years; body mass index 27.5 ± 7.1 kg/m^2^) healthy, non-smoking, experienced divers (minimum certification “Autonomous Divers” according to European norm EN 14153-2 or ISO 24801-2 with at least 50 logged dives) were recruited for this study. All participants held a valid medical clearance for diving at the time of the study. None of them had a history of previous cardiac abnormalities or were under any cardio- or vasoactive medications, nor a history of DCS. They were instructed to abstain from any physical activity and diving for 72 h [6] prior to the experimental protocol. Caffeinated beverages and alcohol were also prohibited for 6 h prior to the test. 

All experimental procedures were conducted in accordance with the Declaration of Helsinki [26] and were approved by the Academic Ethical Committee of Brussels (B200-2020-088) and Ethics Committee Hospital Erasme (P2021/465/B4062021000279).

### 2.2. Study Design

This is a cross-over prospective study. Each diver performed, in a random order, two standardized dives with (experimental condition) and without (control) preconditioning, making each diver his own control. A minimal interval of 1 week between dives was systematically respected. The preconditioning strategy consisted of a 2 min bout of mini trampoline jumping, every diver was asked to perform at least 200 jumps, 15 min before the start of the dive. 

All dives were performed with air, in a pool environment (NEMO 33, Brussels, Belgium) at a depth of 25 m for 25 min. Water and air temperatures were held constant at 33 and 29 °C, respectively, thus needing no thermal protection suit. The descent was carried out at 20 m·min^−1^. At depth, subjects were asked to swim slowly without effort, then come back to the surface with an ascent speed of 10 m·min^−1^. Since this depth time profile falls within accepted “no-decompression limits” [27], no decompression stop was added to the profile.

### 2.3. Bubble Analysis

Although imperfect, it is now accepted that research projects can use VGE data as a surrogate endpoint for decompression stress [9,28]. According to current recommendations, cardiac echography is the gold standard for VGE detection [29]. During field studies, bubbles are usually detected in the right atrium, and graded according to different systems. In this study, during an apical 4 chamber view, echocardiographic VGE signals over the 1 min recording were evaluated by frame-based bubble counting as described by Germonpré et al. [30]. 

Using a two-dimensional echocardiography technique (Sonosite M-Turbo, FUJIFILM Sonosite Inc., Amsterdam, The Netherlands), two measurements were taken at 30 and 60 min post-dive. They were made at rest (without flexion) and following active provocation by two deep knee bends (with flexion). In total, 4 videos of 15 cardiac cycles were recorded for each dive.

At a later stage, these recordings were reviewed to analyze 10 consecutive frames in end-diastolic/protosystolic position and perform a formal bubble counting procedure. Then, the VGE peak count per heartbeat were averaged over these 10 frames, the resulting number was rounded to the closest digit and kept as the final result. The counting was performed independently twice by two trained scientists acquainted with the method used (CB, KL). The numbers of VGE considered for calculation were those that reached consensus.

### 2.4. Flow Mediated Dilation (FMD)

Using a digital diagnostic ultrasound system (DP-30, Mindray, Echomedic, Ghent, Belgium), FMD, an established measure of the endothelium-dependent vasodilation mediated by nitric oxide (NO) [31], was used to assess the effect of diving on main conduit arteries. Brachial artery diameter was measured immediately before and 1 min after a 5 min ischemia induced by inflating a cuff placed on the forearm to 180 mmHg, as previously described [32]. 

All ultrasound assessments were obtained 20–30 min after surfacing while participants rested in the supine position for at least 15 min. All were performed by the same experienced operator (KL) with more than 100 scans/year, which is recommended to maintain competency with the FMD method [33]. 

During image analysis, the brachial artery boundaries were identified manually with an electronic caliper (provided by the ultrasonography software) in a three-fold repetition pattern. The artery diameter was averaged over these three measurements. FMD were calculated as the percent increase in arterial diameter from the resting state to maximal dilation.

### 2.5. Statisitical Analysis

All statistical tests were performed using a standard computer statistical package, GraphPad Prism version 9.00 for MacOS (GraphPad Software, San Diego, CA, USA). 

Normality of data was proven by means of the Kolmogorov–Smirnoff test. Since each diver was his own control, paired tests were mainly used except between both experimental conditions. When a Gaussian distribution was assumed, data were analyzed with a paired *t*-test (or unpaired *t*-test between conditions), and when it could not be assumed, the analysis was performed with a Wilcoxon test (or Mann–Whitney when not paired).

Taking the baseline measures as 100%, FMD changes were calculated for each diving protocol, allowing an appreciation of the magnitude of change rather than the absolute values. Depending on the assumption of the Gaussian distribution, the mean or median of our sample was compared to the hypothetical mean of 100% using either a one-sample *t*-test or a Wilcoxon signed-rank test.

A threshold of *p* < 0.05 was considered statistically significant. All data are presented as mean ± standard deviation (SD). Sample size was calculated setting the power of the study at 90% assuming that variables associated with diving would have been affected to a similar extent to what we already observed in our previous studies [20,25].

## 3. Results

VGE peak count per heartbeat was analyzed for each condition (Figure 1.). No significant difference was found between the 30 and 60 min measurements (control: 2.4 ± 1.9 vs. 3.9 6.8 VGE per heartbeat, *p* = 0.78; mini trampoline: 0.6 ± 1.1 vs. 0.5 ± 1.0 VGE per heartbeat, *p* > 0.99). Therefore, results of both measurements were added to compare conditions. Compared to the control, VGE count was significantly reduced after trampoline preconditioning (control: 3.1 ± 4.9 VGE per heartbeat vs. MT: 0.6 ± 1.1 VGE per heartbeat, *p* = 0.031) (Figure 1). 

Baseline measurements of brachial artery diameter increase before diving were similar in both experimental conditions (control: 0.354 ± 0.06 mm vs. MT: 0.358 ± 0.06 mm; *p* = 0.82). Taking this pre-dive value as the reference (100%), post-dive FMD exhibited a significant decrease in the absence of preconditioning (92.9% ± 7.4 of pre-dive values, *p* = 0.03), which is consistent with previously reported data in the literature [5]. However, mini trampoline preconditioning prevented this post-dive FMD decrease (103.3% ± 7.1 of pre-dive values, *p* = 0.30). The comparison between the experimental conditions was significant (*p* = 0.03). (Figure 2).

## 4. Discussion

VGE start forming during the off-gassing of tissues in the decompression (ascent) phase of a dive and are believed to result from the triggering of bubble precursors (nuclei) into growth. The precise mechanism of micronuclei formation is still debated, with possible sites being located on facilitating endothelial surface regions with surfactants, hydrophobic surfaces or crevices [1,34,35,36]. However, the presumed micronuclei-originated VGE production is consistent with the observation that a certain form of ‘acclimatization’ to decompression stress seems to exist, with a higher probability of VGE for the first dives after a period of non-diving [36,37]. Previous studies have also pointed out a significant inter-subject variability to VGE for the same diving exposure [38]. There is also a large intra-individual variation, indicating that diving time and nitrogen pressure are not the only determinants of VGE formation. 

The limitations associated with VGE grade as a surrogate indicator of decompression stress may explain some possible misinterpretation. Although VGE detection techniques improve (from acoustic Doppler to visual 2D echocardiography to second harmonics echocardiography), there obviously is a size limit below which no VGE will be detected [1,35]. The limits of the sensibility of VGE detection methods imply that small, undetected VGE (less than 22 micrometer [39]) might pass the pulmonary vasculature (which acts as a bubble ‘filter’ [40]) and exert influence in the arterial vascular bed, which could be translated into a modification of arterial ‘stiffness’ as measured by FMD.

Unfortunately, the relative influence of NO-mediated changes (FMD) and VGE have not been satisfactorily clarified in previous literature. Changes in FMD (reduction in FMD post-dive) could modify VGE production (with decreased NO levels acting on micronuclei sites, possibly in crevices or hydrophobic endothelial surface sites) [41,42]. Conversely, decreased VGE production might change FMD by provoking less mechanical or biochemical endothelial reactions. Finally, it is entirely possible that there is no direct relationship between VGE and FMD, but that both are simply caused by the same mechanism through, for instance, a relationship between VGE levels and the presence of endothelial microparticles (MP) [8,43]. Either by shear forces or by mechanical/biochemical damage to the endothelial cell wall, VGE might cause the release of endothelial MP, which will, unlike VGE, readily pass the pulmonary capillary filter.

Although the precise mechanisms by which preconditioning ‘works’ have not yet been determined, several single pre-dive interventions [12,13,20,24] have been demonstrated to be effective to reduce VGE formation and preserve endothelial function. However, previous works from our group tend to demonstrate that there is no direct link between post-dive endothelial function preservation and VGE production. Indeed, some preconditioning strategies, such as whole-body vibration, were very efficient in VGE reduction with lesser action on FMD [12,13], while others such as dark chocolate or red orange complex ingestion before diving [11,44] were very efficient in FMD preservation without a real effect on VGE production. Conversely, sauna preconditioning seemed to be efficient to both reduce VGEs and preserve FMD [20,24], despite the presence of dehydration which should have been promoting VGE [45]. 

Vibration and a sauna might share a common feature, which is a mechanical dislodging of VGE nuclei, either by the transmission of energy shocks or by increased shear forces with increased circulation. Both factors might also be involved in the effects seen by pre-dive exercise preconditioning, as this provokes an increased cardiac output and mechanical movement. In this regard, it is interesting to note that treadmill running seems to have a greater effect than bicycle ergometry [46], probably because of the running-related impacts. In addition to its mechanically related effect, a sauna also improves FMD most probably by an NO-mediated mechanism or by intermediate action through the generation of heat-shock proteins (HSP), both of which are not related to vibration. 

It is interesting to note that MT preconditioning is associated with reduction of 80.4% in VGE production while FMD impairment is prevented (+10.4% compared to control dive). The magnitude of the change in FMD is similar for a sauna, which induced comparable results of 10.9% FMD post-dive [20], while the effect on VGE production is similar to whole-body vibration preconditioning which induced a reduction of 84% in post-dive VGE [12]. Therefore, mini trampoline can be considered a mixed preconditioning strategy which allows the addition of two major actions. Although dive profiles were very close and comparable, the vibration sessions were significantly longer, since they were 30 min long. The MT session was ten times shorter, roughly 2 min to reach the same benefit.

Although the vibration-like effect is far from obvious, a mechanical action cannot be denied. Jumping is associated with blood mass displacement (inertia) causing shear stress while significant muscular contractions and contraction-related ‘squeezing’ of muscular and perimuscular vessels participate in the mechanical dislodging of VGE nuclei. Finally, effort-related compensations associated with jumping on the mini trampoline will activate cardiovascular compensations and putatively heat production as seen in sauna preconditioning. Due to the shorter exposition time, dehydration is probably less likely participating to the effectiveness of this simple method. 

### 4.1. Strengths and Weaknesses

#### 4.1.1. Strengths

−This study builds on established modern methods of evaluation of decompression stress and current theories of VGE generation.−As there is possibly large inter-individual variation for VGE and FMD effects after diving, the subjects served as their own controls.−The measured effects are consistent with the theoretical rationale and do not require complicated new hypotheses.−The equipment used for these experiments is readily available, inviting other research groups to repeat the study.

#### 4.1.2. Weaknesses

−The current study only concerns a small group of subjects.−The subjects were not homogenous or necessarily similar in body composition (age, weight, fat/lean mass distribution, sex).−The subjects were not tested to be ‘consistent bubblers’ before the experiments (this would have required at least three extra identical ‘control’ dives).

## 5. Conclusions

This study indicates that mini trampoline jumping prior to a dive seems to be a very good preconditioning method to reduce VGE and endothelial changes after the dive. It may represent an easy, cheap, and efficient preconditioning for SCUBA diving since it reached comparable benefits as other previously tested methods, in a cost-effective and shorter duration. Larger studies are encouraged to confirm these preliminary results.

## Figures and Tables

**Figure 1 ijerph-19-05410-f001:**
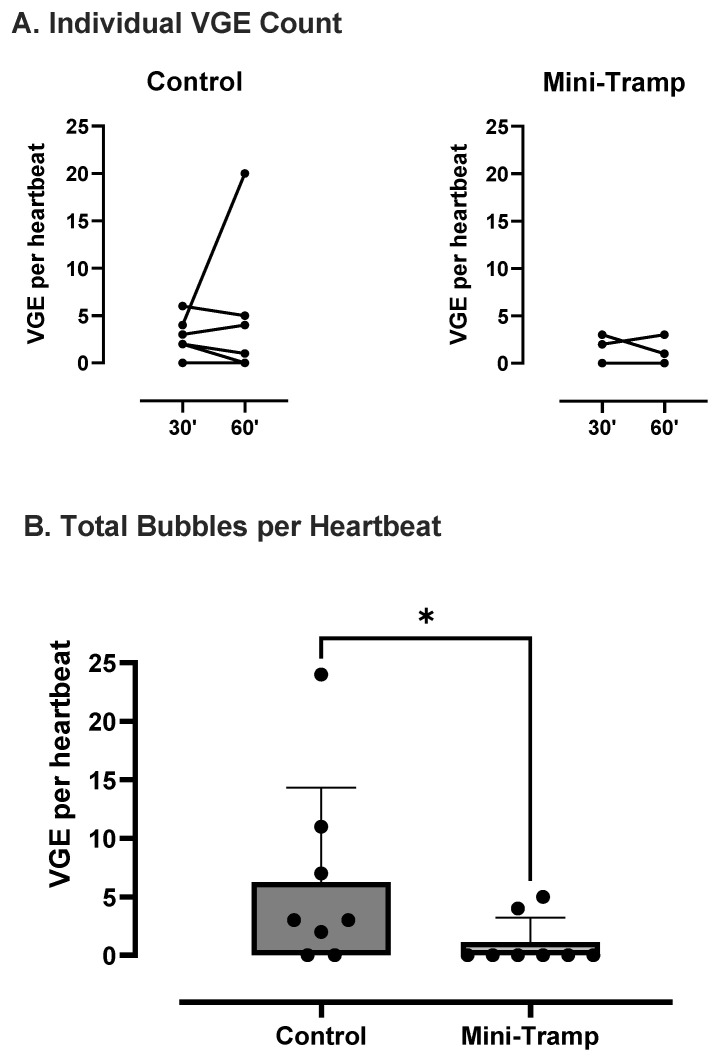
VGE peak count per heartbeat. (**A**) Individual analysis (*n* = 8). (**B**) Total bubble count comparing dives with (mini tramp) and without (control) preconditioning. Each diver is his own control. Data are expressed as means ± SD. (* = *p* < 0.05).

**Figure 2 ijerph-19-05410-f002:**
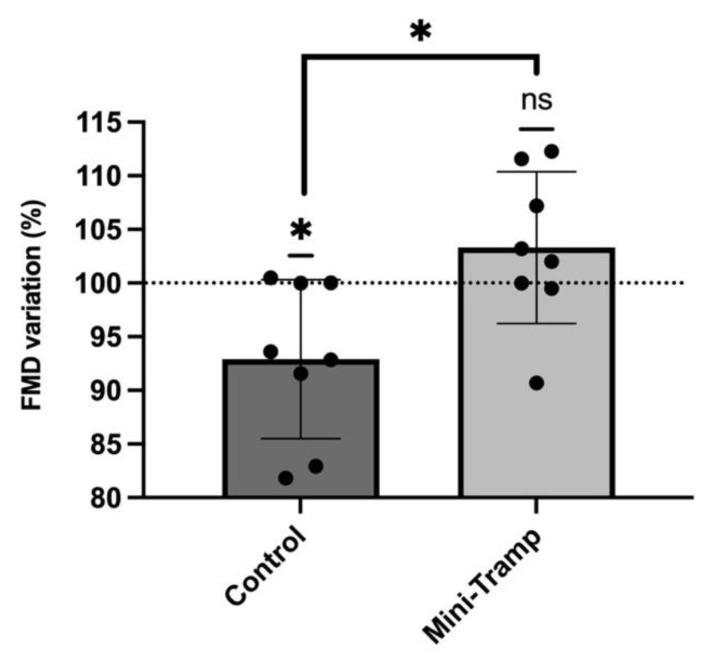
Effects of mini trampoline preconditioning compared to control on endothelial function as measured by FMD. Pre-dive values are taken as reference (100%). Data are expressed as means ± SD. (* = *p* < 0.05; ns = Not Significant).

## Data Availability

The datasets used and analyzed during the current study are available from the corresponding author on reasonable request.

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
