# Peer review of "Mini Trampoline, a New and Promising Way of SCUBA Diving Preconditioning to Reduce Vascular Gas Emboli?"

_ijerph, 2022, doi:10.3390/ijerph19095410_

Round 1

Reviewer 1 Report

This is a very small, but nevertheless interesting study about a new method for preconditioning to reduce VGE in SCUBA-diving.

The overall approach to reduce VGE after diving by eliminating gas nuclei before is not new, but the idea to use a mini trampoline as simple as possible for only 2 minutes is the new aspect of the study.

However, although study design and used methods are sound, I have concerns with data presentation and statistics:

Pooled analysis in fig. 1b: the data from 30 and 60min after diving of one individuum are not independent and thus are not allowed to increase the sample size for statistical calculation. An n=16 gives much more chance of significance than n=8, particular in such a small number.

I have also concerns with fig.2: to set baseline values as reference = 100% eliminates half of the variance in the analysis. As far as I know, such data are not allowed to be tested for significance. I would recommend to use the really measured values only for statistical analysis and not relative changes.

Both figures, results and statistics should be checked by a statistician.

With respect to discussion, after review by a statistician, parts of the results may have lost their significance and need to be discussed in another context.

Ll 213ff: “changes in FMD…” the authors say …with increased NO-levels acting… .

From my knowledge reduction in FMD is caused by reduced NO (due to production of peroxinitrite) – thus the argument “(with increased NO levels acting on micronuclei 214 sites, possibly in crevices or hydrophobic endothelial surface sites)” seems to be not plausible. Please, revise or clarify.

Minor:

L 58: and…

L 62: better “inside…”

L 59: an explanation for the role of knee flexions in bubble testing here? Mobilization of still small and wall adherent gas nuclei/bubbles? Please, explain.

LL 107ff: 2min at a pace of 60-80/min due not fit to at least 200 jumps. Please, correct.

L 124: which position? Apical 4-chamber view? Please, describe.

L 154: better data proof or test…

L 181: artery diameter of 0.3… millimeters? This seems to be the increase in diameter.

Author Response

This is a very small, but nevertheless interesting study about a new method for preconditioning to reduce VGE in SCUBA-diving.

The overall approach to reduce VGE after diving by eliminating gas nuclei before is not new, but the idea to use a mini trampoline as simple as possible for only 2 minutes is the new aspect of the study.

However, although study design and used methods are sound, I have concerns with data presentation and statistics:

Pooled analysis in fig.1b: the data from 30 and 60min after diving of one individuum are not independent and thus are not allowed to increase the sample size for statistical calculation. An n=16 gives much more chance of significance than n=8, particular in such a small number.

We understand the point raised by the reviewer, actually the method used is comparing the total number (mean) of bubbles measured in the two conditions at two time points. This approach is closer to reality, indeed bubbles production after diving is not linear and we are not totally sure with two measurements that we have reached the peak bubble count. Pooling the total number of measured bubbles, we reduce the possibility of having “by chance” targeted peak levels in one of the two compared groups, this approach is potentially less prone to reach statistical levels. In this case we are not comparing the individuals but the bubbles amount. Such approach has been used in several other studies. 

We can agree with the reviewer implementing another approach. Adding the bubbles analyzed for each individual which will also reflect reality as follows: (see corrected figure)

With this approach the calculation is indeed made on n=8, on participants and not on measurements.

I have also concerns with fig.2: to set baseline values as reference = 100% eliminates half of the variance in the analysis. As far as I know, such data are not allowed to be tested for significance. I would recommend to use the really measured values only for statistical analysis and not relative changes.

The FMD measurement is a relative measurement by itself since it is expressed as a percentage of dilatation. Relative measurements are also very commonly used in physiological cross-sectional prospective controlled studies where every subject acts as his/her own control, to avoid discrepant non homogenous baselines. (See referenced articles)

Both figures, results and statistics should be checked by a statistician.

This has been done (statistician corrections) and changes have been made with the same methods used in numerous other studies of our group.

With respect to discussion, after review by a statistician, parts of the results may have lost their significance and need to be discussed in another context.

See previous answers and changes have been made accordingly.

Ll 213ff: “changes in FMD…” the authors say …with increased NO-levels acting… .

From my knowledge reduction in FMD is caused by reduced NO (due to production of peroxinitrite) – thus the argument “(with increased NO levels acting on micronuclei 214 sites, possibly in crevices or hydrophobic endothelial surface sites)” seems to be not plausible. Please, revise or clarify.

 Totally in agreement, thank-you for pointing this, the word should be “decreased”, we corrected accordingly.

Minor:

L 58: and…

Corrected

L 62: better “inside…”

Corrected

L 59: an explanation for the role of knee flexions in bubble testing here? Mobilization of still small and wall adherent gas nuclei/bubbles? Please, explain.

The explanation has been given in previous works on the same kind of subject by the referenced authors: Imbert JP, Egi SM, Germonpre P & Balestra C. (2019). Static Metabolic Bubbles as Precursors of Vascular Gas Emboli During Divers' Decompression: A Hypothesis Explaining Bubbling Variability. Front Physiol 10, 807.

We concur with the reviewer and understand that a little explanation is needed. We added a sentence as follows : ……Doppler or echocardiographic VGE detection is associated with higher bubble release provoked by the muscular contractionsand suggests that this flexion causes still adherent bubbles to be dislodged from their sites before reaching their critical volume of detachment [16]……

LL 107 ff: 2min at a pace of 60-80/min due not fit to at least 200 jumps. Please, correct.

Agree, we simply deleted the frequency mention.

L 124: which position? Apical 4-chamber view? Please, describe. 

Indeed this was an apical four view, we added : “ In this study, during an apical 4 chamber view,.....

L 154: better data proof or test… 

Corrected

L 181: artery diameter of 0.3… millimeters? This seems to be the increase in diameter.

Correct, we changed accordingly adding the word increase as follows:

Baseline measurements of brachial artery diameter increase before diving were..

Thank you for giving us the opportunity to correct ourselves.

Reviewer 2 Report

ijerph-1667123

Mini Trampoline, a new and promising way of SCUBA diving preconditioning to reduce vascular gas emboli?

Summary:

ijerph-1667123 “Mini Trampoline, a new and promising way of SCUBA diving preconditioning to reduce vascular gas emboli?” is a well-presented study of 2 minutes of mini trampoline use to reduce venous gas emboli and maintain flow mediated dilation.

Overall comments:

This is a well-designed and relatively simple study demonstrating that VGA are reduced and FMD is maintained or increased by mini trampoline use.

Specific comments

Page 6, Line 245: Typographical error: Two periods at the end of the sentence, “ . . . dive). . The m

Figures and Tables:

No comments.

References

41 and 49 have alignment formatting issues and are not completely hanging.

Author Response

Mini Trampoline, a new and promising way of SCUBA diving preconditioning to reduce vascular gas emboli?

Summary:

ijerph-1667123 “Mini Trampoline, a new and promising way of SCUBA diving preconditioning to reduce vascular gas emboli?” is a well-presented study of 2 minutes of mini trampoline use to reduce venous gas emboli and maintain flow mediated dilation.

Overall comments:

This is a well-designed and relatively simple study demonstrating that VGA are reduced and FMD is maintained or increased by mini trampoline use.

Thank-you!

Specific comments

Page 6, Line 245: Typographical error: Two periods at the end of the sentence, “ . . . dive). . The m

Corrected

Figures and Tables:

No comments.

References

41 and 49 have alignment formatting issues and are not completely hanging.

Corrected

Reviewer 3 Report

Study absolutely well designed and well written, with a hypothesis of a very intriguing mixed preconditioning procedure (MT) definitely supported by very promising experimental results (with reference to that 80.4% in reduced production of VGE and the prevention of a 10.4% in FMD occurences).
Honest in the analysis of strengths and weaknesses of the study, the latter perhaps are important enough to give reason to a somehow low significance, on my opinion, of the numbers provided by the actual findings. Nevertheless it sounds plausible and feasible that numbers will be confirmed by those other studies following the track given by this innovative one.
Thanks to more homogenous groups, and to the already declared easy and prompt availability of the necessary equipment, these early results are possibly going to lead to their own confirmation, allowing replication and comparison of the data.

Author Response

Study absolutely well designed and well written, with a hypothesis of a very intriguing mixed preconditioning procedure (MT) definitely supported by very promising experimental results (with reference to that 80.4% in reduced production of VGE and the prevention of a 10.4% in FMD occurences).
Honest in the analysis of strengths and weaknesses of the study, the latter perhaps are important enough to give reason to a somehow low significance, on my opinion, of the numbers provided by the actual findings. Nevertheless it sounds plausible and feasible that numbers will be confirmed by those other studies following the track given by this innovative one.
Thanks to more homogenous groups, and to the already declared easy and prompt availability of the necessary equipment, these early results are possibly going to lead to their own confirmation, allowing replication and comparison of the data.

Thank-you!

We added a sentence in the conclusions section encouraging larger studies to confirm our results.
